# Integrative Analysis of miRNA-mRNA in Ovarian Granulosa Cells Treated with Kisspeptin in Tan Sheep

**DOI:** 10.3390/ani12212989

**Published:** 2022-10-30

**Authors:** Tianshu Dai, Xiaolong Kang, Chaoyun Yang, Shan Mei, Shihao Wei, Xingru Guo, Ziming Ma, Yuangang Shi, Yuankui Chu, Xingang Dan

**Affiliations:** 1School of Agriculture, Ningxia University, Yinchuan 750021, China; 2Agriculture and Rural Bureau of Yeji District, Lu’an 237431, China; 3Department of Laboratory Medicine, General Hospital of Ningxia Medical University, Ningxia Medical University, Yinchuan 750004, China

**Keywords:** Tan sheep, granulosa cells, next generation, kisspeptin

## Abstract

**Simple Summary:**

Neurons produce kisspeptin, a peptide hormone that stimulates the pituitary gland to produce gonadotropin and regulate reproductive development. Granulosa cells exist in the ovaries of female animals, secreting hormone receptors and regulating follicular maturation and hormonal balance. The mechanism of regulation of the function of granulosa cells by kisspeptin is still unclear. miRNA-mRNA sequencing was performed on ovarian granulosa cells treated with kisspeptin in Tan sheep to determine the molecular pathways involved. The sequencing results revealed that eight miRNAs significantly differed between the experimental and control groups. The results also indicated that several miRNAs and their target genes regulate steroid production and cell proliferation. This study’s findings will help further explore the molecular mechanism of kisspeptin in the regulation of the function of ovarian granulosa cells in Tan sheep.

**Abstract:**

Kisspeptin is a peptide hormone encoded by the kiss-1 gene that regulates animal reproduction. Our studies revealed that kisspeptin can regulate steroid hormone production and promote cell proliferation in ovarian granulosa cells of Tan sheep, but the mechanism has not yet been fully understood. We speculated that kisspeptin might promote steroid hormone production and cell proliferation by mediating the expression of specific miRNA and mRNA in granulosa cells. Accordingly, after granulosa cells were treated with kisspeptin, the RNA of cells was extracted to construct a cDNA library, and miRNA-mRNA sequencing was performed. Results showed that 1303 expressed genes and 605 expressed miRNAs were identified. Furthermore, eight differentially expressed miRNAs were found, and their target genes were significantly enriched in progesterone synthesis/metabolism, hormone biosynthesis, ovulation cycle, and steroid metabolism regulation. Meanwhile, mRNA was significantly enriched in steroid biosynthesis, IL-17 signaling pathway, and GnRH signaling pathway. Integrative analysis of miRNA-mRNA revealed that the significantly different oar-let-7b targets eight genes, of which EGR1 (early growth response-1) might play a significant role in regulating the function of granulosa cells, and miR-10a regulates lipid metabolism and steroid hormone synthesis by targeting HNRNPD. Additionally, PPI analysis revealed genes that are not miRNA targets but crucial to other biological processes in granulosa cells, implying that kisspeptin may also indirectly regulate granulosa cell function by these pathways. The findings of this work may help understand the molecular mechanism of kisspeptin regulating steroid hormone secretion, cell proliferation, and other physiological functions in ovarian granulosa cells of Tan sheep.

## 1. Introduction

Kisspeptin, a protein encoded by the kiss-1 gene, activates the hypothalamic-pituitary axis to advance puberty and activate gonadotropin (GnRH) and luteinizing hormone (LH) release, promoting follicular development. Kiss-1 and kiss-1 receptors (kiss-1R) are not only expressed in the central nervous system, but also in placenta [1], liver [2], pancreas [2], testis [3,4,5], uterus [2], and ovarian granulosa cells [6]. Ovary-derived kisspeptin regulates follicular development, oocyte maturation, and ovulation by either autocrine or paracrine signaling [7]. Ovarian granulosa cells are closely related to follicular development. Previous studies from this laboratory have shown that kisspeptin could promote progesterone, estrogen secretion, and ovarian granulosa cell proliferation in Tan sheep. However, the specific mechanism has not yet been reported. Studies have shown that miRNA plays a key role in various biological processes and can regulate ovarian granulosa cell proliferation, apoptosis, and steroid hormone secretion [8,9]. The expression of miRNA in tissues is significantly affected by hormones or cytokines [10]. It has been reported that FGF9 treatment increased the expression of miR-221, inhibiting steroid production of ovarian granulosa cells in bovines [11]. Further, follicle-stimulating hormone (FSH) treatment of cultured rat granulosa cells has been reported to affect progesterone synthesis by downregulating the expression of miR-29a and miR-30d and upregulating the expression of miR-23b [12]. It was, therefore, speculated that kisspeptin might affect steroid hormone synthesis and proliferation by altering the expression of key miRNAs in ovarian granulosa cells of Tan sheep.

In recent years, several researchers have identified the expression of miRNA and mRNA in ovarian tissue under various conditions to reveal the molecular regulatory mechanisms of ovarian function using RNA sequencing [13,14]. In this study, we hypothesize that kisspeptin may regulate steroid hormone synthesis and proliferation of ovarian granulosa cells through the kisspeptin-miRNA-mRNA pathway. Thus miRNA-seq and mRNA-seq were performed on ovarian granulosa cells of Tan sheep treated with kisspeptin. The sequencing results were verified by qPCR, followed by the miRNA-mRNA integration analysis, and the miRNA-mRNA interaction relationship was determined. Kisspeptin-mediated miRNA and its targeted regulated mRNA were screened. The signaling pathway regulating steroid production of granulosa cells was further analyzed to determine the molecular mechanism of kisspeptin in the regulation of ovarian granulosa cell function and provide a reference for future studies on the breeding performance of Tan sheep.

## 2. Materials and Methods

### 2.1. Collection of Ovarian Samples

Ovarian samples of Tan Sheep were obtained from Yongning Hongxiang slaughterhouse, Yinchuan, Ningxia. The ovarian tissue was collected from slaughtered female Tan sheep and immediately stored in normal saline containing 1% double antibody at 37 °C. The ovary was brought back to the laboratory within 2 h.

### 2.2. Culture and Treatment of Primary Ovarian Granulosa Cells of Tan Sheep

The collected ovaries were poured into a large beaker containing follicle-washing solution. The connective tissue around the ovaries was cut out with scissors. The ovaries were washed with 75% ethanol solution for 45 s and then rinsed three times for 3 min each in a preheated follicular washing solution. The ovaries were clamped with forceps and 2 mL of DMEM/F12 culture solution was aspirated in a 10 mL syringe. The follicular fluid was extracted from selected 3–6 mm in diameter follicles, placed in DMEM/F12 culture medium tube, and centrifuged at 1000 rpm for 8 min at room temperature. The supernatant was discarded and centrifugation was performed again with a pre-warmed DMEM/F12 culture medium. Then, the second supernatant was discarded and the cells were resuspended in a 3 mL DMEM/F12 culture medium. The cells were evenly distributed in all the wells of a six-well plate. When 70% confluent, the cells were divided into two groups with four replicates in each group. The first group, i.e., the control group (MC group), was cultured by DMEM/F12 medium with sample numbers MC1, MC2, MC3, and MC4. The test group (MT group) was treated with DMEM/F12 containing 500 nM kisspeptin, with sample numbers MT1, MT2, MT3, and MT4. The control and test groups were cultured in 37 °C, 5% CO_2_ concentration cell incubator.

### 2.3. Extraction and Detection of Total RNA from Granulosa Cells

The MC and MT groups were incubated for 24 h as described in the last section. Post-incubation, the six-well plate was removed from the incubator and the culture medium was discarded. The total RNA in the granulosa cells in the MC group (n = 4) and the MT (n = 4) group were extracted by the TRIzol method. The obtained RNA’s integrity was assayed by 1% gel electrophoresis, and RNA concentration and purity were detected using NanoDrop 2000 and Agilent 2100 RNA 6000 Nano kits. The samples with concentrations higher than 500 ng/μL and RNA integrity score ≥ 7 were selected.

### 2.4. Library Building and Sequencing

mRNA libraries, constructed according to the characteristics of mRNA with poly (A) tail, were hybridized with total RNA with poly (T) probe beads to adsorb the mRNA with poly (A) tail. The magnetic beads were recycled to elute poly (A) mRNA from the beads. The eluted mRNA was treated with a magnesium ion solution and then random primers (dNTP) were employed for reverse transcription of the interrupted mRNA fragments to form cDNA. Finally, the “Y” type connector was connected at both ends of the double-stranded cDNA, making a library for hands-on sequencing after PCR amplification. Then, it was used to construct the library in a Small RNA Sample Pre Kit (E7300L, NEB, Ipswich MA, USA). The 5′ end of the small RNA has a phosphate group, while the 3’ end has a hydroxyl group. The linker is directly added to both ends of the small RNA using the total RNA as the starting sample. Next, eight cDNA libraries in the treatment group and the control group were constructed using reverse transcription kits (RR047A, Takara Bio, Dalian, China). cDNA of 350–400 bp were screened, and 2 n mol/L of the library’s effective concentration was needed for precise measurement. The built library was tested using the Agilent 2100 (Agilent 2100, Agilent Technologies, Palo Alto, CA, USA) and the ABI StepOnePlus Real-Time PCR System (ABI StepOnePlus, ABI, CA, USA) for quality and yield. Finally, the Illumina NovaSeq 6000 system (RNA Nano 6000 Assay Kit of the Bioanalyzer 2100, illumina, San Diego, CA, USA) was used to complete the on-machine sequencing of the library according to a standardized process.

### 2.5. Processing and Validation of Sequencing Data

After sequencing, raw reads obtained from transformation were processed to ensure the quality of information analysis by removing reads with joints and low quality in the following steps: The base of the mass value SD ≤ 20 was removed from reads that accounted for more than 30% of the entire read. Reads with N > 10%, 5′ joint contamination, lacking a 3’ joint sequence, and insert fragments, were removed. The 3′ connector sequence was trimmed, and the reads of polyA/T/G/C were removed. DEseq2 [15] analyzed the expression levels of miRNA in the test and control groups. The differential multiple of screening differentially expressed miRNAs and mRNA was ≥1.5 ($\left|\log_{2}\mathrm{FoldChange}\right|\geq 0.585$), *p* < 0.05. Known and unknown miRNAs were identified and annotated using miREvo (Version: miREvo_v1.1, Ming Wen, Guangzhou, China) [16], ViennaRNA (Version: ViennaRNA-2.1.1, Andreas R. Gruber, Wien, Austria) [17], and Mirdeep2 (Version: mirdeep2_0_0_5, Marc R. Friedländer, Berlin-Buch, Germany) [18] databases. Six differentially expressed genes and seven differentially expressed miRNAs were randomly selected for qPCR validation to ensure the transcriptome data sequencing accuracy and reliability. The fluorescent dye, TB Green Primer ExTaqTM II (RR820A, Takara Bio, Dalian, China), was used. The primer information is shown in Table 1. Primer Premier 5 (Version: V5.5.0, Premier, Canada) was used to design the gene primers. Sheep U6 was used as a miRNA internal reference and β-actin as a genetic reference. Novel_621 is a predicted miRNA, so the accession number cannot be found in GenBank. The following reaction conditions were used: 95 °C pre-denaturation for 3 min, 95 °C denaturation for 5 s, 60 °C annealing, and extension for 30 s, 40 cycles.

### 2.6. Kisspeptin-Mediated Analysis of Granulosa Cell Differential mRNA, miRNA Function, and Pathway Enrichment

An enriched analysis of the differential genes was performed based on hypergeometric distribution and BH test principles. GOseq (version number: Release2.12, Matthew D Young, Parkville, Australia) [19] software was used to enrich the differential gene with GO function. The enrichment method was Wallenius. GO analysis included biological processes (BP), cellular components (CC), and molecular function (MF). KOBAS (version number: v2.0, Chen Xie, Beijing, China) [20] software was used to compare sequences with the KEGG (Kyoto Encyclopedia of Genes and Genomes) database sequences and perform KEGG channel enrichment analysis. Enrichment of significant pathways identified the major biochemical metabolic pathways involved in the genes.

### 2.7. Kisspeptin-Mediated Integration Analysis of Granulosa Cells mRNA and miRNA

The differential miRNAs were filtered and compared with the reference genome. The known miRNA and the novel miRNA were annotated and subjected to differential expression analysis. GO and KEGG enrichment analyses were performed on the results to reveal the enrichment of relevant genes in significantly different pathways. MiRanda (version: miRanda-3.3a, Bino John, Massachusetts, USA) [21] and RNAhybrid (version: RNAhybrid v2.0, Jan Krüger, Bielefeld, Germany) [22] online prediction websites were used to predict target genes of miRNAs. The predicted target genes were then intersected with differential mRNA data to analyze miRNA-mRNA pairs with negative regulatory relationships. In order to observe the regulatory relationship between each miRNA and its target genes in the processed samples more intuitively, Cytoscape (version: Cytoscape_v3.8.2, Paul Shannon, California, USA) [23] software was employed to draw the miRNA-mRNA targeting relationship network diagram.

## 3. Results

### 3.1. Cell Culture and Statistical Analysis of Transcriptome Data of Kisspeptin-Treated Granulosa Cells

Tan sheep granulosa cells cultured in vitro for 0 h, 24 h, 48 h, and 72 h are shown in Figure 1. After the data of the control and the test groups were filtered out of the low-quality sequence, the 3′ and 5′ joints were removed. 12,694,192 and 12,995,552 clean reads were obtained, accounting for 92.17% and 98.29%, respectively, of the total reads. More than 97% of sequencing data error rates were lower than Q20 (0.01), and 91% of sequencing data error rates were lower than Q30 (0.001). The sequencing results were, therefore, further analyzed. Sequencing results demonstrated that GC base content in each sample was equal, and the base composition was stable and balanced. An average of 93.81% of clean reads could be compared with the reference genome sequence of sheep (https://www.ensembl.org/index.html, Oar_v3.1, accessed on 25 September 2020). Approximately 72.38% of reads matched a unique location (Table 2). The MC and MT sequences were annotated to the Rfam database for comparison. In the MC, rRNA accounted for 0.75%; snRNA accounted for 0.00%; snoRNA accounted for 0.02%; and tRNA accounted for 0.10%. In the MT, rRNA accounted for 0.11%; snRNA accounted for 0.00%; snoRNA accounted for 0.04%; and tRNA accounted for 0.12% (Table 3). rRNA, snRNA, snoRNA, and tRNA that might exist in samples were found and removed as much as possible.

### 3.2. Identification and Classification Annotation of miRNA

The distribution of the length of small RNA sequences was analyzed. According to the statistical results, most sequences were concentrated in the 18–28 nt, with 21–24 nt sequences having a high frequency. Further, the peak at 23 nt had the highest frequency (Figure 2). According to the sequence abundance statistics of different classifications between the MC and the MT groups, miRNA accounted for 73.57% and 79.51% in the MC and MT group, respectively (Figure 3). About 75% of the sequences were 22–24 nt in length, and the results could be further analyzed.

About 75% of the sequences were 22–24 nt in length and the results were consistent between the two groups.

### 3.3. Differentially Expressed mRNA and miRNA Analysis

There were 1303 differentially expressed genes in the MC and MT groups of ovarian granulosa cells, among which 613 genes were downregulated, and 692 were upregulated. Seven differentially expressed miRNAs were downregulated, and one was found to be upregulated (Figure 4). Further analysis revealed that NPM1 (nucleophosmin 1) and ERH (ERH mRNA splicing and mitosis factor) were significantly downregulated (*p* < 0.05). However, HSBP1 (heat shock factor binding protein 1), RPL35A (ribosomal protein L35a), and other genes were significantly upregulated (*p* < 0.05). Moreover, miR-148a was significantly upregulated (*p* < 0.05), while let-7a, let-7b, let-7c, and miR-10a were significantly downregulated (*p* < 0.05). Cluster analysis heat maps were drawn based on transcription levels in the MC and MT groups (Figure 5). The maps showed that these mRNAs and miRNAs were well clustered and significantly upregulated or downregulated in the samples.

### 3.4. GO Enrichment Annotation of Differentially Expressed mRNA and miRNA

GO function annotation of miRNA-negative related target genes in the kisspeptin-treated test and control group was prepared. According to the results of the GO database, a total of 622 GO terms were annotated. The cellular component was annotated to the endoplasmic reticulum quality control compartment and MHC class I protein complex. The biological process was annotated to regulate the progesterone biosynthetic process and the estrous cycle, and the molecular function was annotated to GTPase activity and zinc ion binding. The most significantly enriched GO entries for mRNA and miRNA are shown in Table 4.

The size and color of the dot represent the number of enriched genes and the magnitude of significance in Figure 6.

### 3.5. KEGG Pathway Analysis of Differential Genes and miRNA Target Genes

KEGG enrichment annotation was performed for the differential target genes predicted by mRNA and miRNA. The enrichment of mRNA results focused on steroid biosynthesis, IL-17 signaling pathway, GnRH signaling pathway, oxytocin signaling pathway, MAPK signaling pathway, ovarian steroidogenesis, and other target pathways (Figure 7). miRNA results were enriched into NF-Kappa B signaling pathway, chemokine signaling pathway, Ras signaling pathway, and PI3K-Akt signaling pathway (Figure 8). 

### 3.6. Kisspeptin-Mediated Key miRNAs for Granulosa Cell Steroid Production and the Corresponding Target Genes

miRNAs are primarily regulated in a negatively correlated manner with the predicted target genes in animals and eukaryotes. In this study, we predicted the target genes of miRNA. Analysis of miRNAs and mRNAs’ associations revealed 16 miRNA-mRNA pairs with negative regulation, forming 22 pairs of targeting relationships. Moreover, the co-expression network of miRNA-mRNA was drawn and it was found that eight miRNAs were in the center of the network, regulating 22 genes (Figure 9), while one miRNA could regulate multiple target genes at the same time. Oar-let-7b could simultaneously regulate TRAF1, MUL1, PTPN23, EGR1, LVRN, ZFHX2, EDEM1, and TUBB2A genes. The EDEM1 gene also simultaneously regulated oar-let-7a, oar-let-7b, oar-let-7c, and oar-let-7d. The gene EGR1, regulating steroid hormone production, targeted the oar-let-7b, while oar-miR-10a targeted the HNRNPD gene. Figure 10 shows the mRNA protein interaction analysis. The interaction diagram between the genes was drawn by selecting the genes with the top 100 confidence levels from the string database.

### 3.7. Validation of Differentially Expressed miRNA and mRNA by qPCR

qPCR analysis of the six differential genes and seven miRNAs revealed that the dissolution curves of all miRNAs and genes were single peak and the primer design was reasonable. The relative expression trends of miRNAs and genes in the MT and MC groups were consistent with the sequencing results of the transcriptome (Figure 11), indicating that the sequencing results were accurate, reliable, and could be used for subsequent functional verification.

As shown in Figure 11, the upregulation and downregulation trends verified by qPCR were precisely the same as those obtained from RNA-seq.

## 4. Discussion

Kisspeptin can regulate animal reproduction by regulating follicular development through the paracrine or autocrine pathway in the gonads. Kisspeptin treatment promotes progesterone secretion in cultured bovine granulosa cells [24], while transfection of the kiss1 gene into porcine ovarian granulosa cells promotes steroid secretion [25]. Our research group also showed that kisspeptin could promote the secretion of steroid hormones and cell proliferation in ovarian granulosa cells of Tan sheep in vitro (unpublished data), but the specific molecular mechanism remains unclear. Recent studies have indicated that kisspeptin-10 could significantly change the expression of circulating miRNAs (let-7e, miR-100-5p, and others) obtained from the plasma of gonads of Senegalese Sole, affecting their reproduction [26]. Kisspeptin-10 has been reported to induce progesterone synthesis in bovine granulosa cells by regulating the expression of miR-146-targeted StAR genes [24]. Therefore, we speculate that kisspeptin affects ovarian granulosa cell function by modulating miRNA and mRNA expression. This study identified some essential mRNA, miRNA, and signaling pathways by integrated analysis after combined sequencing.

GO and KEGG enrichment analysis for miRNAs revealed that 8 of the 31 significant pathways were related to regulating steroid hormones, estrous, cell proliferation, and ovulation. Furthermore, GO and KEGG enrichment analysis for RNA also showed that EGR1, HSD17B12, CYP1B1, etc., were enriched in the regulation of progesterone biosynthesis, estrous cycle, progesterone metabolism, C21-steroid hormone biosynthesis, steroid biosynthesis, ovulation cycle, and steroid metabolism. Meanwhile, twelve upregulated and twelve downregulated genes were also enriched into the PI3K-Akt signaling pathway. It is well known that ovarian granulosa cells primarily express hormone receptors, including follicle-stimulating hormone receptor (FSHR) and luteinizing hormone receptor (LHR), which can regulate ovarian granulosa cell function by binding its ligand. Previous studies have shown that EGR1 regulates embryo implantation [27] and LHR expression [28]. These results suggest that the EGR1 gene may play a specific role in regulating the reproduction of Tan sheep. HSD17B12 belongs to the hydroxysteroid (17β) dehydrogenase family and plays a vital role in female fertility through its role in arachidonic acid (AA) metabolism [29]. CYP1B1, a member of the cytochrome P450 1 subfamily, is mainly responsible for the metabolism of halogenated and polycyclic aromatic hydrocarbons in body tissues. CYP1B1 has been reported to regulate the expressions of progesterone, estrogen, and steroid hormone receptors [30,31,32]. These results indicated that those genes might play significant roles in these pathways regulating the function of ovarian granulosa cells.

Integrative analysis of miRNA-mRNA revealed that the significantly different oar-let-7b targets eight genes, and miR-10a regulates lipid metabolism and steroid hormone synthesis by binding its target gene. Previous studies have discovered that miRNA could regulate follicular development and cell proliferation by binding its target genes [33]. The results of an in vivo ovarian oxidative stress model revealed that downregulation of miR-181a expression inhibited the apoptosis of granulosa cells by upregulating the transcriptional activity of SIRT1 inhibitory pro-apoptotic factors [34]. miR-335-5p has been reported to be involved in granulosa cell proliferation by decreasing SGK3 expression in patients with polycystic ovary syndrome (PCOS) [35]. miR-210 has been reported to regulate the granulosa cell function in the pre-ovulation period through HRA and EFNA3 [36]. All of these suggest that miRNAs play an essential role in regulating granulosa cell function. The let-7 family is the most abundant miRNA detected in the ovary and plays a crucial role in follicles development.

Further, let-7c has been reported to be mainly present in granulosa cells [37]. In this study, four members of the let-7 family (let-7a, let-7b, let-7c, and let-7d) were downregulated. One study demonstrated that let-7b might bind to activin receptorⅠand Smad2/3 genes, affecting follicular development and estrogen secretion through the TGF-β signaling pathway [37]. let-7a was inversely regulated by estrogen and progesterone in the endometrium of estrus mice [38]. Moreover, transfection of let-7b and let-7c can significantly inhibit the release of progesterone from human ovarian granulosa cells [39]. In addition, over-expressed hsa-let-7b could significantly inhibit the proliferation ability of A375 and A2058 cells [40]. Moreover, let-7b might inhibit the proliferation of hepatocellular carcinoma cells [41]. Based on these results, it was speculated that an increase in let-7b levels in vivo might inhibit the expression of target genes associated with steroid hormone secretion, cell proliferation, and follicle development.

In addition, the miR-10 family, including miR-10a and miR-10b, are highly conserved and have similar roles in most species, i.e., inhibiting proliferation and inducing apoptosis of granulosa cells in humans, rats, and mice [39,42]. miR-10a and miR-10b are reported to inhibit proliferation and induce apoptosis of granulosa cells during follicular development by inhibiting BDNF and TGF-β pathways [43]. miR-10a plays an essential role in male germ cell development and spermatogenesis by regulating meiotic processes in male humans and mice [43]. miR-10a-5p overexpression is also reported to reduce the proliferation of porcine ovarian granulosa cells and increase apoptosis. Correspondingly, the transfection of miR-10a-5p inhibitor showed the opposite results [44].

Furthermore, miR-10a reduced the proliferation of granulosa cells in humans [45]. Recent studies have also revealed that the expression of oar-miR-10a in the dominant follicles of sheep was significantly downregulated compared with the pre-follicular recruitment, suggesting that miR-10a affects follicular development by regulating ovarian granulosa cell proliferation [46]. The sequencing results showed that kisspeptin treatment significantly reduced the expression of miR-10a. Therefore, it was speculated that kisspeptin might regulate granulosa cell proliferation and steroid hormone production by downregulating miR-10a in Tan sheep.

Integrative analysis of miRNA-mRNA revealed that HNRNPD, a low-density lipoprotein (LDL) receptor mRNA degradation factor, is a target mRNA of miR-10a. HNRNPD is involved in cholesterol-mediated inhibiting liver LDL receptor expression [47]. Cholesterol is the primary substrate for steroid hormone biosynthesis. Cholesterol uptake from blood lipoprotein particles via LDL receptors is utilized during progesterone and estrogen biosynthesis during follicle development [48]. Hence, HNRNPD may indirectly impact steroid hormone synthesis by regulating LDL receptors. Meanwhile, the knockdown of HNRNPD can inhibit the proliferation of lung cancer cells and glioma cells [49,50]. Therefore, kisspeptin may regulate steroid hormone production and proliferation of ovarian granulosa cells though probably downregulating miR-10a and upregulating HNRNPD. Additionally, EGR1 identified a target mRNA of let-7b by integrative analysis of miRNA-mRNA. EGR1 can regulate follicle development in the ovary, and the knockout of this gene significantly reduces the number of corpus luteum and the levels of progesterone and LHβ, resulting in infertility [51].

Furthermore, one study has shown that EGR1 genes influence cell proliferation and apoptosis [52]. Another study has demonstrated that increased expression of EGR1 could induce cell proliferation by activating the transforming growth factor β1/Smad signaling pathway [53]. As a result, we supposed that kisspeptin might modulate steroid hormone production and proliferation of ovarian granulosa cells by increasing the expression of let-7b and decreasing the expression of EGR1 in Tan sheep.

In addition, in the inter-gene interaction diagram, many genes are not miRNA targets but crucial to other biological processes in granulosa cells. Such as the centrally regulated STC2, TRA2B, and UBE211, of which STC2, known as the estrogen response gene, is expressed in the ovary as a paracrine regulator and is involved in follicular development. STC2 was also reported to promote human follicular development by affecting the proteolytic activity of pregnancy-associated plasma protein A [54]. Accordingly, kisspeptin may also indirectly regulate granulosa cell function by these pathways.

In summary, we identified some key miRNA, mRNA, and important signaling pathways of kisspeptin regulating the function of ovarian granulosa cells by combined sequencing. Integrative analysis of miRNA-mRNA revealed that the significantly different oar-let-7b targets eight genes, of which EGR1 might play a significant role in regulating granulosa cell function, and miR-10a regulates lipid metabolism and steroid hormone synthesis by targeting HNRNPD. However, the precise molecule mechanism of kisspeptin regulating the function of ovarian granulosa cells remains to be investigated in the future.

## 5. Conclusions

This study used RNA-seq techniques to characterize the transcriptome of kisspeptin-treated and untreated ovarian granulosa cells of Tan sheep. Further, key miRNAs (miR-10a, let-7b, and let-7c) and genes (EGR1, HNRNPD) regulating steroid hormone production and cell proliferation were screened, and the direct involvement of the targeted relationship (let-7b-EGR1, miR-10a-HNRNPD) in regulating the function of granulosa cells was discovered by integrative analysis of miRNA-mRNA. PPI analysis revealed genes there are not miRNA targets but are crucial to other biological processes in granulosa cells. The findings of this work may help understand the molecular mechanism of kisspeptin regulating steroid hormone secretion, cell proliferation, and other physiological functions in ovarian granulosa cells of Tan sheep.

## Figures and Tables

**Figure 1 animals-12-02989-f001:**
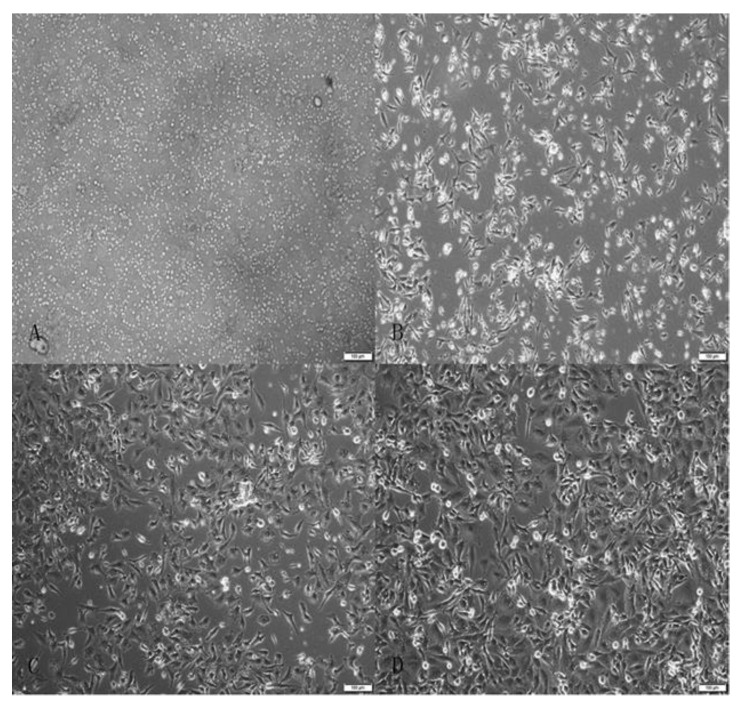
In vitro cultivation of granulosa cells. (**A**) Freshly isolated granulosa cells. (**B**) Granulosa cells cultured for 24 h. (**C**) Granulosa cells cultured for 48 h. (**D**) Granulosa cells cultured for 72 h.

**Figure 2 animals-12-02989-f002:**
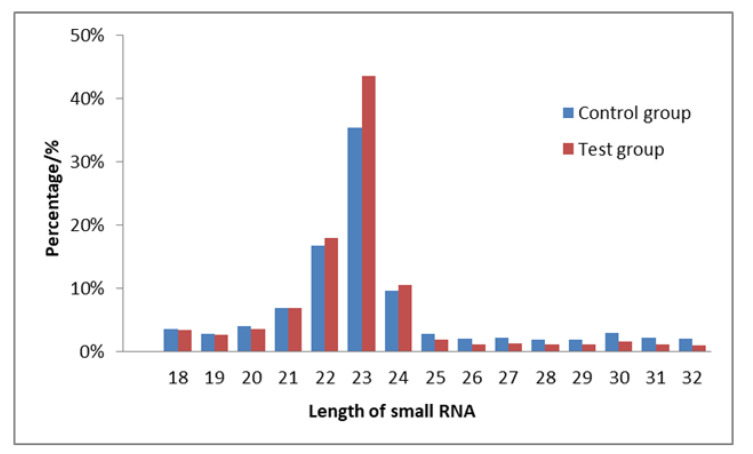
Distribution of small RNA length in the MC and the MT groups.

**Figure 3 animals-12-02989-f003:**
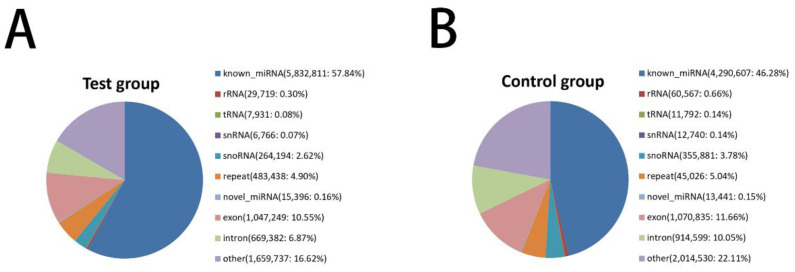
Small RNA classification notes. (**A**) Classification notes for test groups. (**B**) Classification notes for control groups.

**Figure 4 animals-12-02989-f004:**
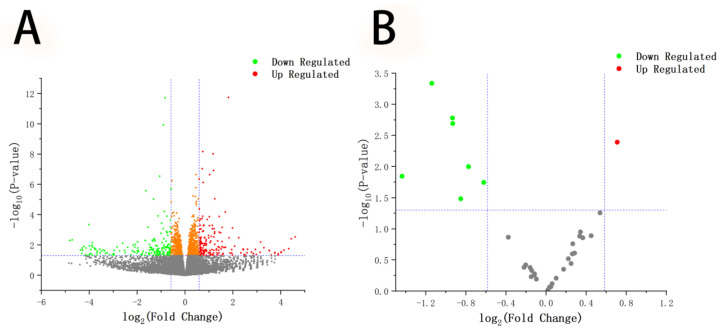
Volcanic map of differentially expressed mRNA and miRNA. The volcanic map shows the number of differentially expressed genes and miRNAs in the control and test groups. Red and green represent the upregulated and downregulated genes and miRNAs.

**Figure 5 animals-12-02989-f005:**
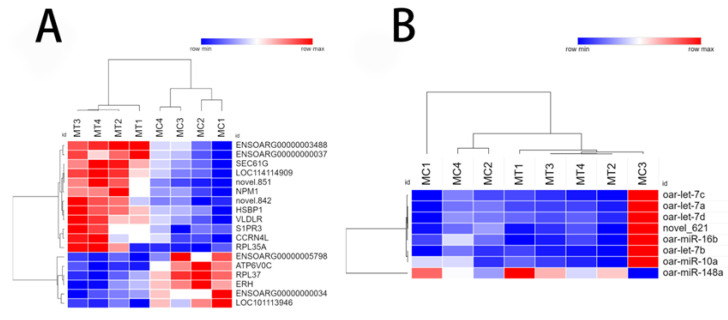
Heat map of differentially expressed mRNA (**A**) and heat map of differentially expressed miRNA (**B**). Significantly different genes and miRNAs clustering heat map. Red and blue represent upregulated and downregulated, respectively.

**Figure 6 animals-12-02989-f006:**
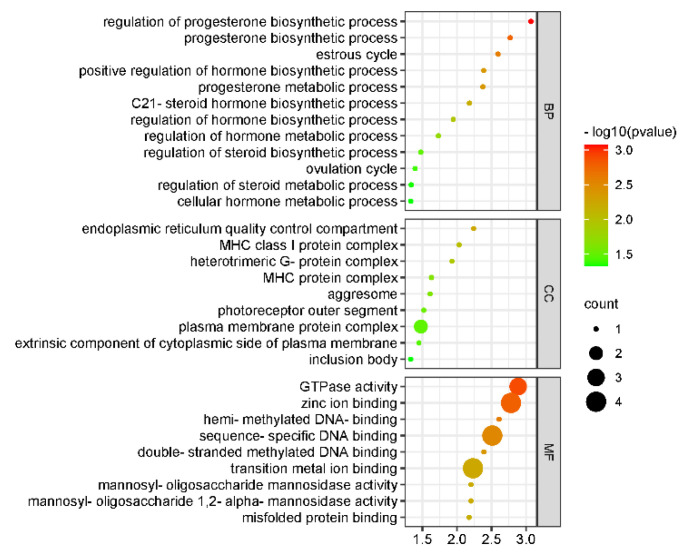
GO analysis of miRNA.

**Figure 7 animals-12-02989-f007:**
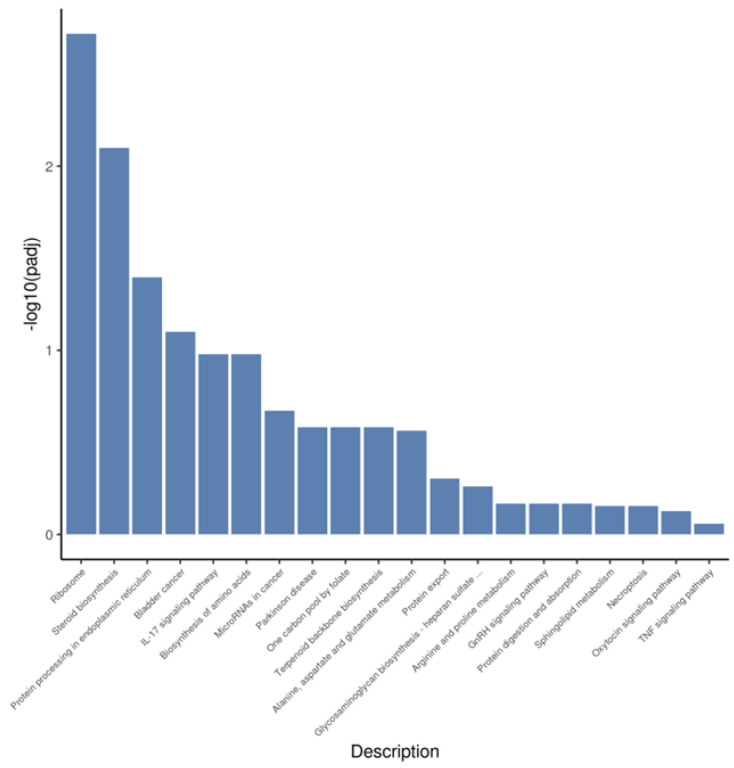
KEGG enrichment pathway of significantly different genes. Different column heights represent the magnitude of enrichment significance of this pathway.

**Figure 8 animals-12-02989-f008:**
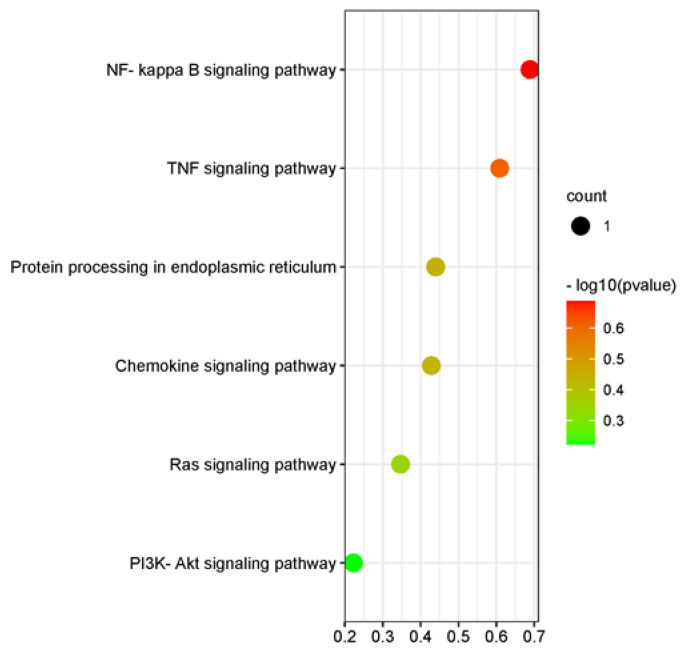
A significant difference in the target gene KEGG pathway of miRNA. The color of the dot represents the magnitude of significance in Figure 8.

**Figure 9 animals-12-02989-f009:**
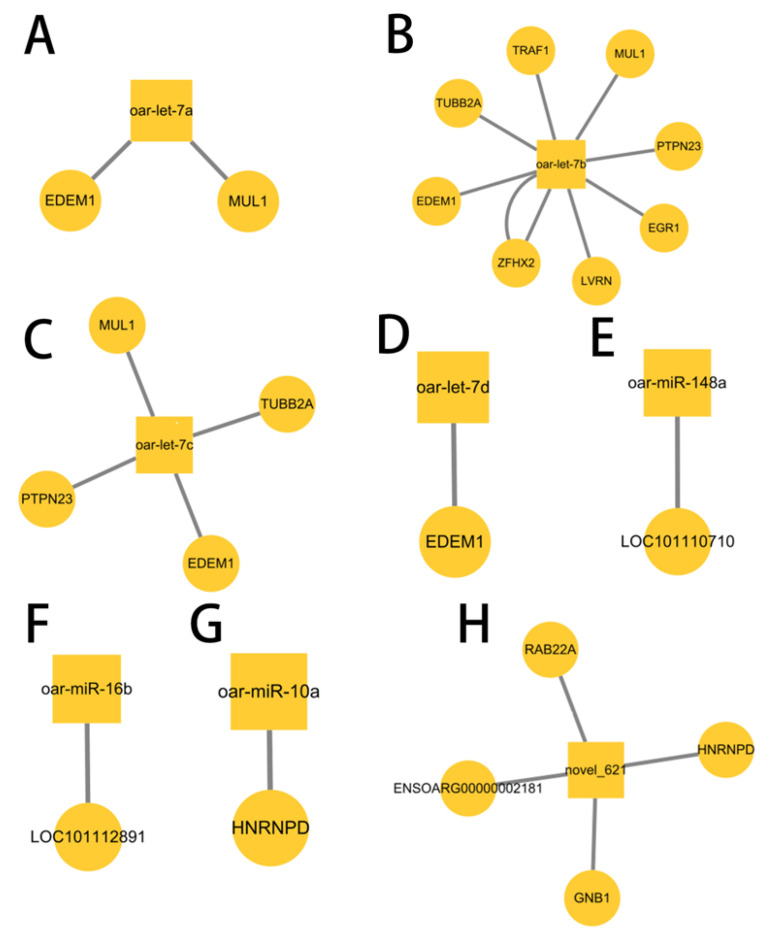
miRNA-mRNA co-expression network diagram. Squares represent miRNAs, circles represent mRNAs, and lines represent interactions. It can be seen from the figure that one miRNA can regulate multiple genes and a single gene can also regulate multiple miRNAs.

**Figure 10 animals-12-02989-f010:**
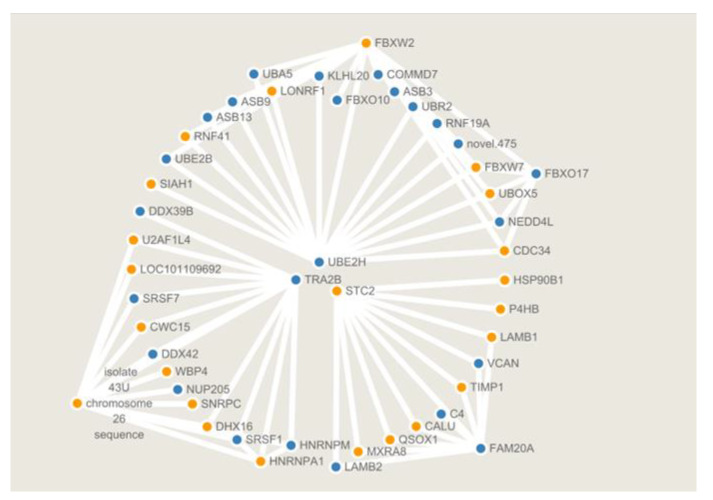
Interaction relationship between functionally related genes and target genes. Each node represents a different gene. Orange nodes represent the upregulation of the gene and blue nodes represent the downregulation of the gene. The line represents the existing interaction relationship. It can be seen intuitively from the figure that genes such as UBE2H, TRA2B, STC2, FBXO17, and FBXW2 have critical regulatory functions in this interaction network.

**Figure 11 animals-12-02989-f011:**
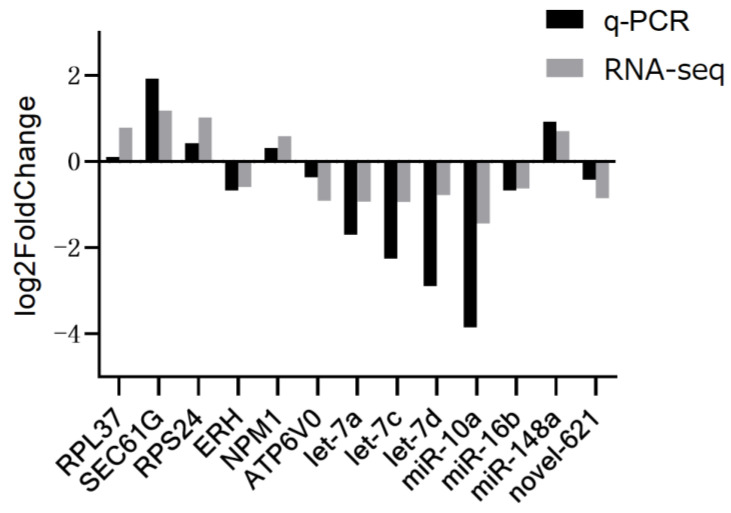
qPCR and RNA-seq quantitative analysis of differentially expressed genes and miRNA.

**Table 1 animals-12-02989-t001:** qPCR primer sequences.

Name	GenBank	Sequence (5′→3′)	T_m_/°C	Length/bp
β-actin	KU365062.1	F: GGCATTCACGAAACTACCTTC R: ATCTCTTTCTGCATCCTGTCTG	60 °C	134
RPL37	XM_004017021.1	F: GGGAACGTCATCGTTTGG R: TGCCTGAATCTGCGGTAT	60 °C	219
ATP6V0C	NM_001009195.1	F: CCTACGGGACAGCCAAGAG R: TGAGGACTGCCACCACCAG	60 °C	135
ERH	XM_004010751.4	F: CGCTGACTATGAATCTGT R: GCTGGTATGTCTGGGTAT	60 °C	185
SEC61G	XR_006056971.1	F: GGGCGTCTGTCGGCTCTTGT R: GCGGCTCTGAGTCAGCTTTCC	60 °C	181
NPM1	XM_027979982.2	F: CTGGAGCAAAGGATGAGT R: TCAAAGTCGCCAGTGTTA	60 °C	87
RPS24	XM_004021510.4	F: TGGCTTCGGCATGATTTA R: GTTCTTGCGTTCCTTTCG	60 °C	127
U6	NR_138085.1	F: CTCGCTTCGGCAGCACA R: AACGCTTCACGAATTTGCGT	60 °C	94
oar-miR-16b	NR_107949.1	F: CGGGCTAGCAGCACGTAA R: CAGCCACAAAAGAGCACAAT	60 °C	65
oar-let-7c	HE600003.1	F: CGGGCTGAGGTAGTAGGTTG R: CAGCCACAAAAGAGCACAAT	60 °C	65
oar-let-7a	HE599994.1	F: CGGGCTGAGGTAGTAGGTTG R: CAGCCACAAAAGAGCACAAT	60 °C	65
oar-miR-148a	HE599922.1	F: CGGGCTCAGTGCACTACAGA R: CAGCCACAAAAGAGCACAAT	60 °C	65
oar-let-7d	HE599918.1	F: CGGGCAGAGGTAGTAGGT R: CAGCCACAAAAGAGCACAAT	60 °C	65
oar-miR-10a	HE599840.1	F: CGGGCTACCCTGTAGATCC R: CAGCCACAAAAGAGCACAAT	60 °C	65
novel_621	--	F: CGGGCCAGGATTAACCAGAGG R: CAGCCACAAAAGAGCACAAT	60 °C	65

**Table 2 animals-12-02989-t002:** Gene comparison analysis after quality control.

Sample	Total Mapped	Multiple Mapped	Uniquely Mapped
MC1	49,047,802 (84.32%)	3,107,292 (5.34%)	45,940,510 (78.98%)
MC2	57,264,742 (89.5%)	3,741,509 (5.85%)	53,523,233 (83.66%)
MC3	65,128,932 (89.63%)	4,434,166 (6.1%)	60,694,766 (83.52%)
MC4	55,498,882 (90.15%)	3,800,007 (6.17%)	51,698,875 (83.98%)
MT1	56,304,168 (90.55%)	3,989,488 (6.42%)	52,314,680 (84.13%)
MT2	55,612,293 (89.91%)	3,790,133 (6.13%)	51,822,160 (83.78%)
MT3	53,071,271 (89.53%)	3,392,251 (5.72%)	49,679,020 (83.81%)
MT4	54,125,070 (88.98%)	3,394,546 (5.58%)	50,730,524 (83.4%)

Note: MC is the control group; MT is the test group.

**Table 3 animals-12-02989-t003:** The tag abundance statistics of non-coding RNA in Rfam were compared between MC and MT groups.

Sample	Total	rRNA	snRNA	snoRNA	tRNA
MC	1,763,919	1315 (0.75%)	94 (0.00%)	298 (0.02%)	1801 (0.10%)
MT	1,234,441	1379 (0.11%)	102 (0.00%)	450 (0.04%)	2207 (0.12%)

**Table 4 animals-12-02989-t004:** The most significant GO entries for differentially expressed genes and miRNAs.

GO Accession	Term Type	GO Entries	*p*-Value	Type
GO:0043603	Biological process	Cellular amide metabolic process	6.11 × 10^−8^	DEGs
GO:0043604		Amide biosynthetic process	7.36 × 10^−8^	
GO:0006518		Peptide metabolic process	2.02 × 10^−7^	
GO:0043043		Peptide biosynthetic process	3.63 × 10^−7^	
GO:0006412		Translation	7.89 × 10^−7^	
GO:1901566		Organo nitrogen compound biosynthetic process	6.00 × 10^−6^	
GO:0005840	Cellular component	Ribosome	5.10 × 10^−7^	
GO:1990904		Ribonucleoprotein complex	2.09 × 10^−6^	
GO:0043228		Non-membrane-bounded organelle	1.01 × 10^−5^	
GO:0043232		Intracellular Non-membrane-bounded organelle	1.01 × 10^−5^	
GO:0036510	Biological process	Trimming of terminal mannose on C branch	8.28 × 10^−4^	DEmiRNAs
GO:0098758		Response to interleukin-8	8.47 × 10^−4^	
GO:0098759		Cellular response to interleukin-8	8.47 × 10^−4^	
GO:2000182		Regulation of progesterone biosynthetic process	8.47 × 10^−4^	
GO:0097167		Circadian regulation of translation	1.69 × 10^−3^	
GO:0006701		Progesterone biosynthetic process	1.69 × 10^−3^	
GO:0060086		Circadian temperature homeostasis	1.69 × 10^−3^	
GO:0097494		Regulation of vesicle size	2.30 × 10^−3^	
GO:0003924	Molecular function	GTPase activity	1.30 × 10^−3^	
GO:0008270		Zinc ion binding	1.65 × 10^−3^	

## Data Availability

All raw data during the current study are available in the NCBI BioProject (https://submit.ncbi.nlm.nih.gov/subs/bioproject) with accession number PRJNA895403 (accessed on 30 September 2022).

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
