# Peer review of "Integrative Analysis of miRNA-mRNA in Ovarian Granulosa Cells Treated with Kisspeptin in Tan Sheep"

_animals, 2022, doi:10.3390/ani12212989_

Round 1
Reviewer 1 Report
The paper is interesting. The aim of this study is to explore the mechanisms that regulate kisspeptin effects upon granulosa cells.
However, the discussion is very difficult to read. Is a long list of data present in the literature but it is not easy to understand the correlation with the data presented in this paper.
One minor point in the discussion
"Parietal granulosa cells primarily secrete hormone receptors, including FSHR and LHR"
Granulosa cells EXPRESS gonadotropin receptors do not secrete them
Author Response
Point 1: The discussion is very difficult to read. Is a long list of data present in the literature but it is not easy to understand the correlation with the data presented in this paper.
Response 1: Thank you for the reviwer’s kind suggestion. We have made change the discussion and further associated with the article data. The discussion is as follows:
- Discussion
Kisspeptin can regulate animal reproduction and participate in the regulation of follicular development through paracrine or autocrine pathway in gonad. Kisspeptin treatment promotes progesterone secretion in cultured bovine granulosa cells [23], while transfection of kiss1 gene into porcine ovarian granulosa cells promotes steroid secretion [24]. Our research group also showed that kisspeptin could promote the secretion of steroid hormones and cell proliferation in ovarian granulosa cells of Tan sheep in vitro (unpublished data), but the specific molecular mechanism remains unclear. Recent studies have indicated that kisspeptin-10 could significantly change the expression of circulating miRNAs (let-7e, miR-100-5p, and others) obtained from the plasma of gonads of Senegalese Sole, affecting their reproduction [25]. Kisspeptin-10 has been reported to induce progesterone synthesis in bovine granulosa cells by regulating the expression of miR-146-targeted StAR genes [23]. Therefore, we speculate that kissppetin affect the function of ovarian granulosa cells though probably modulating the expression of miRNA and mRNA. In this study, we identified some key mRNA, miRNA and signaling pathway by the integrated analysis after combined sequencing.
GO and KEGG enrichment analysis for miRNAs revealed that 8 of the 31 significant pathways were related to regulating steroid hormones, estrous, cell proliferation and ovulation. Furthermore, GO and KEGG enrichment analysis for RNA also showed that EGR1, HSD17B12, CYP1B1, et al, were enriched in the regulation of progesterone biosynthesis, estrous cycle, progesterone metabolism, C21-steroid hormone biosynthesis, steroid biosynthesis, ovulation cycle, and steroid metabolism. Meanwhile, twelve up-regulated and twelve down-regulated genes were also enriched into the PI3K-Akt signaling pathway. It is well known that ovarian granulosa cells primarily express hormone receptors, including follicle-stimulating hormone receptor (FSHR) and luteinizing hormone receptor (LHR), which can regulate the function of ovarian granulosa cells by binding its ligand. Previous studies have shown that EGR1 regulates embryo implantation [26] and LHR expression [27]. These results suggest that the EGR1 gene may play a specific role in regulating the reproduction of Tan sheep. HSD17B12 belongs to the hydroxysteroid (17β) dehydrogenase family and plays a vital role in female fertility through its role in arachidonic acid (AA) metabolism [28]. CYP1B1, a member of the cytochrome P450 1 subfamily, is primarily responsible for the metabolism of halogenated and polycyclic aromatic hydrocarbons in body tissues. CYP1B1 has been reported to regulate the expressions of progesterone, estrogen, and steroid hormone receptor [29-31].These results indicated that those genes may play major roles in these pathways regulating the function of ovarian granulosa cells.
Integrative analysis of miRNA-mRNA revealed that the significantly different oar-let-7b targets 8 genes, and miR-10a regulates lipid metabolism and steroid hormone synthesis by binding its target gene. Previous studies have discovered that miRNA could regulate follicular development and cell proliferation by binding its target genes [32]. The results of an in vivo ovarian oxidative stress model revealed that down-regulation of miR-181a expression inhibited the apoptosis of granulosa cells by up-regulating the transcriptional activity of SIRT1 inhibitory pro-apoptotic factors [33]. miR-335-5p has been reported to be involved in granulosa cell proliferation by decreasing SGK3 expression in patients with polycystic ovary syndrome (PCOS) [34]. miR-210 has been reported to regulate the granulosa cell function in the pre-ovulation period through HRA and EFNA3 [35]. All of these suggest that miRNAs play an important role in the regulation of granulosa cell function. Let-7 family is the most abundant miRNA detected in the ovary and plays a crucial role in follicles development. Further, let-7c has been reported to be mainly present in granulosa cells [36]. In this study, four members of the let-7 family (let-7a, let-7b, let-7c, and let-7d) were down-regulated. One study demonstated that let-7b might bind to activin receptor I and Smad2/3 genes, affecting follicular development and estrogen secretion through the TGF-β signaling pathway [36]. Let-7a was inversely regulated by estrogen and progesterone in the endometrium of estrus mice [37]. Moreover, transfection of let-7b and let-7c can significantly inhibit the release of progesterone from human ovarian granulosa cells [38]. Additionally, over-expressed hsa-let-7b could significantly inhibit the proliferation ability of A375 and A2058 cells [39]. Also, let-7b might inhibit the proliferation of hepatocellular carcinoma cells [40]. Based on these results, it was speculated that an increase in let-7b levels in vivo might inhibit the expression of target genes associated with steroid hormone secretion, cell proliferation and follicle development.
In addition, the miR-10 family, including miR-10a and miR-10b, are highly conserved and have similar roles in most species, i.e., inhibiting proliferation and inducing apoptosis of granulosa cells in humans, rats, and mice [38,41]. miR-10a and miR-10b are reported to inhibit proliferation and induce apoptosis of granulosa cells during follicular development by inhibiting BDNF and TGF-β pathways [42]. miR-10a plays an essential role in male germ cell development and spermatogenesis by regulating meiotic processes in male humans and mice [42]. Overexpression of miR-10a-5p is also reported to reduce the proliferation of porcine ovarian granulosa cells and increase apoptosis. Correspondingly, the transfection of miR-10a-5p inhibitor showed the opposite results [43]. Furthermore, miR-10a reduced the proliferation of granulosa cells in human [44]. Recent studies have also revealed that the expression of oar-miR-10a in the dominant follicles of sheep was significantly down-regulated compared with the pre-follicular recruitment, suggesting that miR-10a affect follicular development by regulating ovarian granulosa cell proliferation [45]. The sequencing results showed that kisspeptin treatment significantly reduced the expression of miR-10a. Therefore, it was speculated that kisspeptin might regulate proliferation and steroid hormones production of granulosa cells by down-regulating miR-10a in Tan sheep.
Integrative analysis of miRNA-mRNA revealed that HNRNPD, a low-density lipoprotein (LDL) receptor mRNA degradation factor, is a target mRNA of miR-10a. HNRNPD is involved in cholesterol-mediated inhibition of liver LDL receptor expression 46]. Cholesterol is the primary substrate for steroid hormone biosynthesis. Cholesterol uptake from blood lipoprotein particles via LDL receptors is utilized during progesterone and estrogen biosynthesis during follicles development [47]. Hence, HNRNPD may indirectly impact steroid hormone synthesis by regulating LDL receptors. Meanwhile, knockdown of HNRNPD can inhibit the proliferation of lung cancer cells and glioma cells [48,49]. Therefore, kisspeptin may regulate steroid hormone production and proliferation of ovarian granulosa cells though probably down-regulating miR-10a and up-regulating HNRNPD. Additionally, early growth response-1 (EGR1) was identified a target mRNA of let-7b by integrative analysis of miRNA-mRNA. EGR1 can regulate follicle development in the ovary, and the knockout of this gene significantly reduces the number of corpus luteum and the levels of progesterone and LHβ, resulting in infertility [50]. Furthermore, one study has been shown to EGR1 genes influencing proliferation and apoptosis of cells [51]. Another study has demonstrated that increased the expression of EGR1 could induce cell proliferation by activating transforming growth factor β1/Smad signaling pathway [52]. As a result, we supposed that kisspeptin may modulate steriod hormones production and proliferation of ovarian granulosa cells by increasing the expression of let-7b and decreasing the expression of EGR1 in Tan sheep.
In addition, in the inter-gene interaction diagram, there are many genes that are not miRNA targets but are crucial to other biological processes in granulosa cells. Such as the centrally regulated STC2, TRA2B, UBE211, of which STC2, known as the estrogen response gene, is expressed in the ovary as a paracrine regulator and is involved in follicular development. STC2 was also reported to promote human follicular development by affecting the proteolytic activity of pregnancy-associated plasma protein A [53]. Accordingly, kisspeptin may also indirectly regulating granulosa cell function by these pathways.
In summary, we identify some key miRNA, mRNA and important signaling pathways of kisspeptin regulating the function of ovarian granulosa cells by combined sequencing. Integrative analysis of miRNA-mRNA revealed that the significantly different oar-let-7b targets 8 genes, of which EGR1 might play major role in regulating the function of granulosa cells, and miR-10a regulates lipid metabolism and steroid hormone synthesis by targeting HNRNPD. However, the precise molecule mechanism of kisspeptin regulating the function of ovarian granulosa cells remains to be investigated in future.
Point 2: One minor point in the discussion
"Parietal granulosa cells primarily secrete hormone receptors, including FSHR and LHR"
Granulosa cells EXPRESS gonadotropin receptors do not secrete them.
Response 2: We are very sorry for our incorrect expressing. We have changed the word "secrete" to "express" in line 365.
Reviewer 2 Report
Manuscirpt ID: animals-1943026
Title: Integrative Analysis of miRNA-mRNA in Ovarian Granulosa Cells Treated with Kisspeptin in Tan Sheep
In this manuscript, the authors implemented miRNA and mRNA sequencing on ovarian granulosa cells treated with kisspeptin in Tan sheep to determine the molecular pathways involved. The authors found that several key miRNAs and genes regulate reproductive cycle and steroid, and confirmed their target relationship in the estrus process. The result is interesting and might provide useful information for exploring the molecular mechanism of kisspeptin in the regulation of the function of ovarian granulosa cells in sheep. However, in section 2.2 and results, the authors didn’t poof the obtain cells are ovarian granulosa cells instead of other cells. This is very import for further analysis. In additon, the picture of ovarian granulosa cells is missing. This lead the potential readers might doubt the reliability of the result. In section 2.5,2.6 and 2.7, the authors didn’t cite any reference when using a software. This is CANNOT be acceptable for any scientific journal. Thus, this manuscript should be rejected and encourage resubmission after revisions.
Author Response
Point 1: However, in section2.2 and results, the authors didn’t proof the obtain cells are ovarin granulosa cells instead of other cells. This is very important for further analysis. In attion, the picture of ovarian granulosa cells is missing.
Response 1: Thanks for the referee’s suggestion. In fact, the identification of ovarian granulosa cells had already been done by another graduated student applying the FSHR immunofluorescence in Tan sheep (as shown in following Figure). On the basis of his research, we further explored the mechanism of kisspeptin regulating steroidogenic in ovarian granulosa cells of Tan sheep. In our study, the methods for collecting ovarian samples, isolating and culturing ovarian granulosa cells were identical to those used by the graduate student, ensuring that the cells isolated and cultured by us were ovarian granulosa cells of Tan sheep. Also, we have also provided his master's thesis, and identification of ovarian granulosa cells in Tan sheep by FSHR immunofluorescence was detailed in page11of Chapter 2. Meanwhile, we have also added our own graphs at 0h, 24h, 48h, 72h after culturing ovarian granulosa cells (Figure 1), which, as you mentioned, provides a more convincing result.
(Please refer to the pictures in Response to 2 Comments)
Figure. Expression of FSHR in ovarian granulosa cells of Tan sheep cultured in vitro
A1: FSHR staining in control group; A2: Nuclear staining in control group; A: a combination of A1 and A2; B1: FSHR staining in the experimental group; B2: Nuclear staining in the experimental group; B: The combination of B1 and B2.
(Please refer to the pictures in Response to 2 Comments)
Figure 1. In vitro cultivation of granulosa cells
(A) Freshly isolated granulosa cells. (B) Granulosa cells cultured for 24 h. (C) Granulosa cells cultured for 48 h. (D) Granulosa cells cultured for 72h.
Point 2: In section2.5, 2.6 and 2.7, the authors didn’t cite any reference when using a software.
Response 2: Thanks for the referee’s reminder. We have corrected our misunderstanding in using the software, and we have added references to the software in text. Thank you for your valuable comments again.
[15] Wen, M.; Shen, Y.; Shi, S.H.; Tang, T. miREvo: an integrative microRNA evolutionary analysis platform for next-generation sequencing experiments. BMC Bioinformatics. 2012, 13(1), 140-140.
[16] Lorenz, R.; Bernhart, S.H.; Siederdissen, C.H.; Tafer, H.; Flamm, C.; Stadler, P.F.; Hofacker, I.L. ViennaRNA Package 2.0. Algorithms Mol Biol. 2011, 6(1), 26.
[17] Friedlander, M. R.; Mackowiak, S. D.; Li, N. Chen, W.; Rajewsky, N. miRDeep2 accurately identifies known and hundreds of novel microRNA genes in seven animal clades. Nucleic Acids Res. 2012, 40(1), 37-52.
[18] Young, M.D.; Wakefield, M.J.; Smyth, G.K.; Oshlack, A. Gene ontology analysis for RNA-seq: accounting for selection bias. Genome Biol. 2010, 11(2), R14.
[19] Xie, C.; Mao, X.; Huang, J.; Huang, J.J.; Ding, Y.; Wu, J.M.; Dong, S.; Kong, L.; Gao, G.; Li, C.Y.; et al. KOBAS 2.0: a web server for annotation and identification of enriched pathways and diseases. Nucleic Acids Res. 2011, 39, W316-W322.
[20] Bino, J.; Enright, A.J.; Aravin, A.; Tuschl, T.; Sander, C.; Marks, D.S. Human MicroRNA Targets. PLoS Biol. 2004, 2(11), e363.
[21] Kruger, J.; Rehmsmeier, M. RNAhybrid: microRNA target prediction easy, fast and flexible. Nucleic Acids Res. 2006, 34, 451-454.
[22] Shannon, P.; Markiel, A.; Ozier, O.; Baliga, N.S.; Wang, J.T.; Ramage, D.; Amin, N.; Schwikowski, B.; Ideker, T. Cytoscape: a software environment for integrated models of biomolecular interaction networks. Genome Res. 2003, 13(11), 2498-2504.

Reviewer 3 Report
This is an interesting manuscript about the study of the molecular mechanisms involved in the regulation of the function of granulosa cells by kisspeptin. However, there are several grammar suggestions that should be considered to improve the quality of the manuscript:
General comments:
- The Abstract should be a total of about 200 words maximum; currently the abstract is 370 words in length.
- In the introduction section, I suggest to include some references about previous studies of RNA-sequencing (i.e., mRNA) related to ovarian tissues.
- Also suggest to include in the Introduction section the hypothesis being tested in the study.
- In the text, citation references in square brackets should be separated from the text by a space; please correct it throughout the manuscript.
- In Materials and Methods section, units of measure should be separated from the quantity.
- In References section, all references should be corrected following the guidelines described in the “Instructions for Authors”.
- First part of the manuscript is single-spaced, whereas second part is double-spaced.
Minor grammar comments:
- Line 136: Replace semicolon sign by a period sign.
- Line 155: Replace “Analysis” by “analysis”.
- Line 158: Insert the words “was used” at the end of the row.
- Line 160: Replace “Analysis” by “analysis”.
- Line 178: Replace “Analysis” by “analysis”.
- Line 205: Separate the words “frequency(Figure 1)”.
- Line 207: Separate the words “respectively(Figure 2)”.
- Line 219: The line must be justified,
- Line 221: The line must be justified.
- Line 221: Replace “Analysis” by “analysis”.
- Lines 222-224: Separate round brackets from gene names by a space.
- Line 269: Remove the period sign before the round bracket.
- Lines 281-285: The names of the software (i.e., miRanda-3.3a and Cytoscape) were already included in Materials and Methods section.
- Line 401: Remove the word “cholesterol” because it’s duplicated.
- Line 432: Remove the word “While”.
Author Response
Point 1: The Abstract should be a total of about 200 words maximum; currently the abstract is 370 words in length.
Response 1: Thanks for your kind suggestion. We have reduced the abstract to 252 words,as follows:“Kisspeptin is a peptide hormone encoded by the kiss-1 gene that regulates animal reproduction. Our studies have revealed that kisspeptin can regulate steroid hormone production and promote cell proliferation in ovarian granulosa cells of Tan sheep, but the mechanism has not yet been fully understood. We speculated that kisspeptin might promote steroid hormone production and cell proliferation by mediating the expression of specific miRNA and mRNA in granulosa cells. Accordingly, after granulosa cells were treated with kisspeptin, the RNA of cells was extracted to construct cDNA library, and miRNA-mRNA sequencing were performed. Results showed that a total of 1303 expressed genes and 605 expressed miRNAs were identified. Furthermore, eight differentially expressed miRNAs were found, and their target genes were significantly enriched in progesterone synthesis/metabolism, hormone biosynthesis, ovulation cycle, and steroid metabolism regulation. Meanwhile, mRNA was significantly enriched in steroid biosynthesis, IL-17 signaling pathway, GnRH signaling pathway, et al. Integrative analysis of miRNA-mRNA revealed that the significantly different oar-let-7b targets 8 genes, of which EGR1 might play major role in regulating the function of granulosa cells, and miR-10a regulates lipid metabolism and steroid hormone synthesis by targeting HNRNPD. Additionally, PPI analysis revealed genes there are not miRNA targets but are crucial to other biological processes in granulosa cells, implying that kisspeptin may also indirectly regulating granulosa cell function by these pathways. The findings of this work may be useful in understanding the molecular mechanism of kisspeptin regulating steroid hormone secretion, cell proliferation and other physiological functions in ovarian granulosa cells of Tan sheep.”
Point 2: In the introduction section, I suggest to include some references about previous studies of RNA-sequencing (i.e., mRNA) related to ovarian tissues.
Response 2: Thanks for your kind reminder. We have added two references of RNA-sequencing related to ovarian tissue in the text.
[13]. Zhao, Z.Q.; Wang, L.J.; Sun, X.W.; Zhang, J.J.; Zhao, Y.J.; Na, R.S.; Zhang, J.H. Transcriptome Analysis of the Capra hircus Ovary. PLoS One. 2015, 10(3): e121586.
[14]. Song, P.Y.; Yue, Q.X.; Fu, Q.; Li, X.Y.; Li, X.J.; Zhou, R.Y.; Chen, X.Y.; Tao, C.Y. Integrated analysis of miRNA–mRNA interaction in ovaries of Turpan Black Sheep during follicular and luteal phases. Reprod Domest Anim. 2021, 56(1), 46-57.
Point 3: Also suggest to include in the Introduction section the hypothesis being tested in the study.
Response 3: Thanks for your kind reminder. We have added the hypothesis of this study in the last paragraph of the introduction, as follows: “In recent years, several researchers have identified the expression of miRNA and mRNA in ovarian tissue under various conditions in order to reveal the molecular regulatory mechanisms of ovarian function using RNA sequencing [13,14]. In this study, we hypothesis that kisspeptin may regulate steroid hormone synthesis and proliferation of ovarian granulosa cells though kisspeptin-miRNA-mRNA pathway, thus miRNA-seq and mRNA-seq were performed on ovarian granulosa cells of Tan sheep treated with kisspeptin. The sequencing results were verified by qPCR followed by the miRNA-mRNA integration analysis, and the miRNA-mRNA interaction relationship was determined. Kisspeptin mediated miRNA and its targeted regulated mRNA were screened. The signaling pathway regulating steroid production of granulosa cells was further analyzed to determine the molecular mechanism of kisspeptin in the regulation of ovarian granulosa cell function and provide a reference for future studies on the breeding performance of Tan sheep.”
Point 4: In the text, citation references in square brackets should be separated from the text by a space; please correct it throughout the manuscript.
Response 4: We are very sorry for this mistake. We have separated the citation references from the text with a space in the whole manuscript.
Point 5: In Materials and Methods section, units of measure should be separated from the quantity.
Response 5: Thanks for your kind reminder. We have separated the units of measure form the quantity in Meterials and Methods section.
Point 6: In References section, all references should be corrected following the guidelines described in the “Instructions for Authors”.
Response 6: Thanks for your kind reminder. We have corrected all references following the guidelines described in the “Instructions for Authors”.
Point 7: First part of the manuscript is single-spaced, whereas second part is double-spaced.
Response 7: Thanks for your kind reminder. We have set the spacing between lines of text to 0.95 following the guidelines described in the “Instructions for Authors”.
Point 8: Line 136: Replace semicolon sign by a period sign.
Response 8: Thanks for your kind reminder. We have replaced the semicolon sign by period sign in line 162.
Point 9: Line 155: Replace “Analysis” by “analysis”.
Response 9: Thanks for your kind reminder. We have replaced the “Analysis” by “analysis” in line 179.
Point 10: Line 158: Insert the words “was used” at the end of the row.
Response 10: Thanks for your kind reminder. We have inserted the words “was used” at the end of the row in line 183.
Point 11: Line 160: Replace “Analysis” by “analysis”.
Response 11: Thanks for your kind reminder. We have replaced the “Analysis” by “analysis” in line 184.
Point 12: Line 178: Replace “Analysis” by “analysis”.
Response 12: Thanks for your kind reminder. We have replaced the “Analysis” by “analysis” in line 203
Point 13: Line 205: Separate the words “frequency(Figure 1)”.
Response 13: Thanks for your kind reminder. We have separated the words “frequency (Figure 1)” in line 232.
Point 14: Line 207: Separate the words “respectively (Figure 2)”.
Response 14: Thanks for your kind reminder. We have separated the words “respectively (Figure 2)” in 235.
Point 15: Line 219: The line must be justified.
Response 15: Thanks for your kind reminder. We have justified the line 246.
Point 16: Line 221: The line must be justified.
Response 16: Thanks for your kind reminder. We have justified the line 248.
Point 17: Line 221: Replace “Analysis” by “analysis”.
Response 17: Thanks for your kind reminder. We have replaced the “Analysis” by “analysis” in line 248.
Point 18: Lines 222-224: Separate round brackets from gene names by a space.
Response 18: Thanks for your kind reminder. We have separated round brackets from gene names by a space in lines 248-251.
Point 19: Line 269: Remove the period sign before the round bracket.
Response 19: Thanks for your kind reminder. We have removed the period sign before the round bracket in line 289.
Point 20: Lines 281-285: The names of the software (i.e., miRanda-3.3a and Cytoscape) were already included in Materials and Methods section.
Response 20: Thanks for your kind reminder. We have removed these to avoid duplication in line 299-303.
Point 21: Line 401: Remove the word “cholesterol” because it’s duplicated.
Response 21: Thanks for your kind reminder. We have deleted the word “cholesterol” and rewritten the discussion section of the manuscript and further associated with the article data according to first reviewer’s suggestion.
Point 22: Line 432: Remove the word “While”.
Response 22: Thanks for your kind reminder. We have deleted the word “While” and rewritten the discussion and further associated with the article data according to first reviewer’s suggestion.

Round 2
Reviewer 1 Report
This version of the paper has been improved and I think that can be accepted
Author Response
Point 1:This version of the paper has been improved and I think that can be accepted.
Response 1:Thank you again for all your suggestions for this article.
Reviewer 2 Report
The authors addressed many of my major concerns. However, some issues be addressed before accept for publication.
L316 Please put this result in Results Section.
L121,L125,L127-128, L145 Please provide cat. Number, company, city, postcode and country information.
L138 “DEseq2”, Please add a reference.
Some English grammar errors should be checked carefully. For example, “Tan sheep granulosa cells cultured in vitro for 0, 24, 48, 72 hours are shown (Figure 1).” can be revised as “... are shown in Figure 1”
